# The Association between Parenting Stress, Positive Reappraisal Coping, and Quality of Life in Parents with Autism Spectrum Disorder (ASD) Children: A Systematic Review

**DOI:** 10.3390/healthcare10010052

**Published:** 2021-12-28

**Authors:** Mein-Woei Suen, Valendriyani Ningrum, Muhamad Salis Yuniardi, Nida Hasanati, Jui-Hsing Wang

**Affiliations:** 1Psychology Faculty, University of Muhammadiyah Malang, Jl. Raya Tlogo Mas No. 246, Malang 65141, Jawa Timur, Indonesia; zahroh@umm.ac.id (N.); salis@umm.ac.id (M.S.Y.); nida@umm.ac.id (N.H.); 2Department of Healthcare Administration Specialty in Psychology, College of Medical and Health Science, Asia University, Taiwan No. 500, Liufeng Road, Wufeng District, Taichung City 41354, Taiwan; 3Department of Psychology, College of Medical and Health Science, Asia University, Taiwan No. 500, Liufeng Road, Wufeng District, Taichung City 41354, Taiwan; 4Gender Equality Education and Research Center, Asia University, Taiwan No. 500, Liufeng Road, Wufeng District, Taichung City 41354, Taiwan; 5Department of Medical Research, Asia University Hospital, Asia University, Taiwan No. 500, Liufeng Road, Wufeng District, Taichung City 41354, Taiwan; 6Department of Medical Research, China Medical University Hospital, China Medical University, Taiwan No. 91, Xueshi Road, North District, Taichung City 404, Taiwan; 7Department of Preventive and Public Health Dentistry, Faculty of Dentistry, Baiturrahmah University, Jalan Raya Bypass km 15 Air Pacah, Koto Tangah, Padang 25586, Sumatera Barat, Indonesia; valend888@gmail.com; 8Department of Management, Faculty of Economics and Business, University of Muhammadiyah Malang, Jl. Raya Tlogo Mas No. 246, Malang 65141, Jawa Timur, Indonesia; widayat@umm.ac.id; 9Department of Infection, Taichung Tzu Chi Hospital, Buddhist Tzu Chi Medical Foundation, Taichung City 427, Taiwan

**Keywords:** parenting stress, positive reappraisal coping, quality of life, parents with ASD children

## Abstract

Parents with autism spectrum disorder (ASD) children generally suffer from poor coping and an impaired quality of life (QoL). This systematic review investigates parenting stress, positive reappraisal coping, and QoL in parents with ASD children. A literature search was carried out for publications written in English on the selected investigation topics using five databases, namely, Scopus, PubMed, Wiley, ScienceDirect Online, and EBSCO. Only studies investigating or measuring parenting stress, positive reappraisal coping, and QoL in ASD were included. Our results indicate that parents with ASD children have high stress levels. This is associated with the ineffective use of coping strategies and a low QoL. Adaptive coping strategies are related to a higher QoL, whereas maladaptive coping techniques are related to a worse QoL. Positive reappraisal coping is negatively correlated to meaningfulness, and it is used by parents to change their daily routines in order to motivate themselves towards new and evolving goals in life. Finally, we found a significant negative correlation between parenting stress, positive reappraisal coping, and the QoL of parents with ASD children. Positive reappraisal coping as a strategy helps parents adapt to stress and was found to be a potential mediatory function between parental stress and QoL.

## 1. Introduction

An autism spectrum disorder (ASD) is a developmental condition identified by medical criteria such as deficiencies in social communication and social interaction, and the existence of limited, repeated patterns of behavior, interests, or activities that may continue throughout life [1]. Parents with ASD children are key members of their ASD children’s health teams, and serve as the primary nurses for their ASD children [2]. They may encounter more caregiving issues than parents of typical children, such as higher costs of therapy, childcare difficulties, locating therapeutic facilities that are reasonably priced due to a lack of clinical resources and governmental assistance [3], and preserving their socioeconomic status [4]. These problems can detrimentally impact the care of children with ASD, particularly when these problems are coupled with parental health issues such as stress [5], anxiety disorders [6], and quality of life (QoL) [7]. In addition, their inability to meet parenting responsibilities can affect the parents’ physiological and cognitive function and, in conjunction with changes in their behavior, may prevent them from performing their parental duties appropriately, and, eventually, lead to ASD children feeling more negative emotions [5,7,8].

Many studies demonstrate that parents with ASD children have more stress [9,10,11,12,13], use more emotion-focused coping, which is not actually an ineffective coping strategy, [14], and a worse QoL (19). Understanding how they manage these difficult conditions, cope with the stress, and maintain a good quality of life is an interesting topic. Moreover, the connection between factors affecting the well-being of parents with ASD children is also not completely understood yet [15]. 

Previous studies partially investigate parenting stress, positive reappraisal coping, and quality of life. For instance, some studies have sought to use various broad understandings of how communities engage with stressful events to discover coping strategies [16,17], whereas others have only investigated how using a particular method can be better when it comes to successful stress management and well-being [16,17,18]. The difficulty in understanding how parenting stress and positive reappraisal coping correlate to the quality of life of parents with ASD children stems from a lack of agreement in the literature on how these are defined and quantified. The research focus of this study is the possible correlation between parenting stress, positive reappraisal coping strategies, and the QoL of parents with ASD children. To the best of our knowledge, there is no previous study investigating all these variables comprehensively, and it is the first thorough investigation evaluating the link between parenting stress, coping strategy, and QoL in parents with ASD children. Secondly, it also aims to investigate whether positive reappraisal coping serves as a mediator between parental stress and QoL.

Examining the correlation between these three variables and the mediator role of the positive reappraisal coping will be beneficial, particularly in terms of the stress management improvement of parents with ASD children. In addition, the effective coping strategies expected to be found by this study could be a worthy recommendation for the better QoL management of parents with ASD children. This study aims to investigate the association between parenting stress, positive reappraisal coping, and QoL in parents with autism spectrum disorder (ASD) children. This is an important topic to explore, because improving the quality of life of parents with ASD children is crucial in maintaining their stress and mental health.

## 2. Materials and Methods

### 2.1. Literature Search Strategy 

In this study, a literature search was carried out for publications written in English from five databases—Scopus, PubMed, Wiley, ScienceDirect Online (SDOL), and EBSCO—, and the study was then published following the Preferred Reporting Items for Systematic Reviews and Meta-Analyses (PRISMA) checklist. The research was limited to the latest research from 1 November 2010 to 1 November 2020 to ensure that the research was modern and acceptable. The research population was included in the search strategy, which included terms and keywords acquired through scoping searches as well as existing knowledge in the topic area. A thesaurus software was employed to determine the terms for searching articles. Search words were used to find articles that addressed the search strategies’ three primary concerns: (1) autism disorder or autism spectrum disorder, (2) parenting stress or stress of parents with ASD children, (3) coping strategies or positive reappraisal coping, and (4) quality of life or well-being. Appendix A details the search approach utilized in each of the electronic databases. Duplicates were eliminated from the results when they were exported to EndNote. All references were entered into EndNote, which is specialized software for handling bibliographies. This systematic review was not registered in advance.

### 2.2. Eligibility Criteria 

To be included in our inquiry, the article had to have reported on the influence of parenting stress on positive reappraisal coping, and the QoL of parents with ASD children. The studies must also have included a correlational study design, been written in English, and used clinical and/or non-clinical samples. To ensure that all studies included in this review were peer-reviewed for scientific rigor, articles needed to have used a case series or quality design, whereas abstracts, books, proceedings, oral/poster presentations, systematic reviews, meta-analyses, and studies that did not report on parenting stress or positive reappraisal were excluded. This limited our review to studies that included information about ASD children to better understand parenting stress, positive reappraisal coping, and QoL.

### 2.3. Analytical Process 

At each stage of the review process, two reviewers (N.Z. and V.N.) separately searched information sources and assessed the discovered studies’ eligibility for inclusion using preset criteria. A study’s whole text was examined. For abstracts with inadequate information, or in cases of conflict, the full text was requested. When both reviewers independently determined that a study met the inclusion criteria after examining the complete text, it was included. The risk of prejudice in each experiment was separately evaluated by the same examiners. In cases of conflict, both reviewers mediated a decision. Cohen’s criterion was used to arrive at an agreement between reviewers. Before being used, all tools and processes were tested. The Joanna Briggs Institute critical evaluation checklist was used to examine the risk of bias [19]. The data were then presented in tabular form to enable quantitative and qualitative comparisons of parenting stress, the positive reappraisal coping style, and QoL amidst parents with ASD children.

### 2.4. Study Selection

The relevance of each article was first determined based on the data supplied in the abstract. A reproduction of the complete text was requested after the abstract failed to give sufficient information. The two reviewers independently assessed the abstracts using our search criteria and then chose publications for full-text review. Articles with full text were evaluated for eligibility using preset criteria. Studies had to use an observational study design to explore parenting stress, positive reappraisal coping, and the quality of life related to the well-being of parents with ASD children to be included in the final analysis.

### 2.5. Quality Assessment

The kappa statistic was used to compare the results of these reviews to determine the amount of agreement between the two reviewers. Studies were considered if the kappa score for the selected study was more than 0.91, as determined by both reviewers. “Fair” agreement was defined as kappa values between 0.40 and 0.59, “good” agreement was defined as kappa values between 0.60 to 0.74, and “excellent” agreement was defined as kappa values of 0.75 or above [20]. When the two reviewers’ scores for an article diverged, a consensus score was assigned following a thorough discussion.

### 2.6. Data Extraction and Synthesis

Data were extracted using a Microsoft Excel sheet (Microsoft Corporation, Redmond, WA, USA) in accordance with the data extraction template developed by the Cochrane Consumers and Communication Review Group. The Excel sheet was used to examine inclusion criteria and then tested for all the papers that were chosen. The two authors worked independently on this process. Any disagreements about study eligibility were settled during their conversations. The reasons for excluding full-text articles were recorded in the Excel sheet. 

The risk of bias was appraised using the ROBINS-E tool (University of Bristol, Bristol, UK). The ROBINS-E technique was developed by an international team of investigators and has several advantages, including offering a formal and transparent method for assessing bias risk in observational research [20]. The ROBINS-E tool can assess risks of bias such as confounding, the selection of participants in the study, the classification of exposures, departures from intended exposures, missing data, the measurement of outcomes, and the selection of the reported result. For each of these items, signaling questions were posed to assist the user in determining decisions. Finally, the judgments within each domain were combined to create an overall risk of bias rating for each study [20]. The risk of bias was assessed independently by the same two authors.

## 3. Results

### 3.1. Search and Screening

During this literature search, 5043 articles were identified from Scopus (n = 554), PubMed (n = 455), Wiley (n = 780), ScienceDirect Online (SDOL) (n = 1781), and EBSCO (n = 1473). The citations were entered into a review management program, and 15 duplicates were eliminated. A total of 149 quotes was found for the full-text examination after the initial screening phase, 4 of which were duplicates. A total of 120 articles was excluded during the second screening process. Finally, 29 papers were chosen for peer review. Figure 1 depicts the research selection procedures in greater detail.

Most of the eliminated studies met more than one exclusion criterion. A total of 24 articles covered populations over the age of 18; 24 studies on pediatric disorders were deemed ineligible for this investigation; 4 research studies were conducted in a language other than English; 26 were abstracts only; finally, ineligible study designs were used in 16 of the papers (e.g., analysis reviews, meta-analyses, experimental designs, and description analyses that did not address the relationship between coping, parenting, and QoL) and 26 studies did not satisfy the parenting stress and QoL criteria. Therefore, this review included a total of 29 articles for a further analysis (see Table 1). 

### 3.2. Characteristics of the Research Included

There were 29 studies included, all of which were cross-sectional studies investigating ASD (see Table 1). The number of studies published by different geographical areas was as follows: the United States (12), Australia (1), China (2), India (3), Europe (6), Taiwan (1), Hong Kong (1), Malaysia (1), Pakistan (1), and Jordan (1). The review sample included a total of 8392 parents. In nearly all the included studies, most research respondents were mothers.

### 3.3. Instruments for Measuring Main Constructs

#### 3.3.1. Parenting Stress Measurement

To measure parenting stress, several instruments were utilized (see Table 2). Eleven research studies employed the Parenting Stress Index-Short Form (PSI-SF) [15,21,22,23,24,25,26,27,28,29,30]. The PSI-SF is a self-report measure to evaluate relationships between the stress of parents and children. There are twenty elements directly related to parents in the self-rating depression scale (SDS) to evaluate a parent’s level of stress. Ashworth et al. [31] used the GSSS (Genetic Syndromes Stressors Scale) to quantify parental stressors in relation to unusual genetic illnesses with 14 items. One study used the 2016 National Survey of Children’s Health (NSCH) to assess family resilience [32], and included three stress-related items to measure parenting stress and four new items to assess family resilience. In addition, Rodriguez et al. [33] used the Burden Interview to assess psychological anguish and difficulty when parenting and caring for children. Two studies [34,35] used the PRSS (Perceived Stress Scale), which was developed to assess perceived stress in mothers. 

Tomeny [36] employed the Questionnaire on Resources and Stress-Short Form (QRS-F) for evaluating parental adaptability and coping with developmental limitations in raising a child with physical disabilities or a chronic illness. Costa et al. [37] used the heart rate variability (HRV) index, since individual differences in cardiac activity (HRV) represent an objective sign of the brain’s ability to organize autonomous emotional responses of the nervous system. Cappe et al. [38] used the ALES (Appraisal of Life Events Scale). The Questionnaire on Resources and Stress-Short Form for families with chronic diseases or disabilities was used by Pisula and Porebowicz-Dorsmann [39], whereas the DASS-21 (Depression, Anxiety, and Stress Scale-21) was used by Seymour et al. [40].

#### 3.3.2. Measurement of Positive Reappraisal Coping 

To assess caregiver coping techniques, five self-reported questionnaires were used (Table 2). Begum et al. [41] and Pisula and Kossakowska [42] utilized an updated version of a coping checklist. It was separated into eight subgroups: confrontative, self-control, distancing, searching for support networks, admitting authority, avoiding escape, problem solving planning, and positive reassessment. In addition, four studies used the Brief-COPE (Coping Orientation to Problems Experienced) [15,26,40,43]. Cappe et al. [44] and Cappe, Bolduc, Rougé, Saiag, and Delorme [38] used the French version of the WCC-R (Ways of Coping Checklist) to evaluate parents’ techniques for coping with stress caused by their children. Ishtiaq et al. [45] used the Coping Strategies Inventory Short Form (CSI-SF) to evaluate stress levels and parent coping techniques.

#### 3.3.3. Measurement of Quality of Life (QoL)

To evaluate QoL, Pisula and Porebowicz-Dorsmann [39] employed the WHOQO-BREF (World Health Organization Quality of Life Assessment Questionnaire) (see Table 2). It consists of 26 items and assesses physical and mental health, and social and environmental relationships. Hsiao [46] used a family quality of life scale to measure and evaluate parental satisfaction and QoL. It comprised 25 questions divided into five categories (family, emotional well-being, parenting, physical/material well-being, and support for disability). In addition, Chu et al. [47] measured the overall QoL in terms of happiness using the CarerQol visual analog scale (CarerQol-VAS) and the CarerQol-7D. Bohadana, Morrissey and Paynter [22] assessed the effects and repercussions of ASD on many dimensions of daily life using a quality of life of ASD scale. Furthermore, Nuske et al. [48] used the Pediatric Quality of Life (PedsQL) inventory to measure the occurrence of issues in several domains of functioning over the previous four weeks, such as physical, social, emotional, and cognitive functioning. 

### 3.4. Evaluation of Quality

We evaluated the quality of the selected studies using the Joanna Briggs Institute’s checklist for analytical cross-sectional studies. The checklist was created using information from the study design, study execution, and analysis of the desired outcome. Studies of high quality obtained seven to eight marks, while those of moderate quality received four to six. Studies with a point total of less than four were disqualified. The listed studies were evaluated by two writers (N.Z. and V.N.).

### 3.5. Parenting Stress and Factors Related to Parenting Stress

Previous research indicated that parents with ASD children experience the highest amount of parenting stress compared to parents of children with mental retardation, cerebral palsy, and typically developed children [23,26,30,34]. Thus, the research demonstrated that parents with ASD children reported more stress and depression (see Table 3). Another study discovered that mothers with ASD children had more stress in parenting than mothers with cerebral palsy children [25,34]. Interestingly, there were gender differences for depression in parents. For example, Gong, Du, Li, Zhang, An, and Wu [23] found that mothers with ASD children were more likely to suffer from depression than fathers, although the stress levels of fathers were found to have increased by Rivard, Terroux, Parent-Boursier, and Mercier [29]. Thus, emotional differences between parents of different genders are a necessary area for further investigation. Among the 22 research articles included in the parenting stress review, 1 study discovered at least some indication that certain factors, such as parents’ negative evaluation of their ASD children’s lability, were significantly predictive of parental stress [37]. One study reported that the behavior problems and emotional impairments of the child are predictors of parenting stress [25]. Other studies found that predictors of parenting stress are age, behavioral problems, and anxiety [23]. Furthermore, parental self-efficacy lowers parental stress substantially [27,32].

### 3.6. Association between Parenting Stress and Positive Reappraisal Coping 

As previously discussed, (see Table 3), the data indicated that parents with ASD children reported increased stress and depressive symptoms [26,45]. According to Begum, Islam, and Rahman [41], this disorder requires parents to attempt to handle their situation with various methods based on their abilities. However, Hall and Graff [24] discovered that while parents can manage externalizing behaviors, they often do not have the tools or the awareness necessary to manage internalizing behaviors. Problems with child behavior may lead to parental exhaustion, which may result in the use of inadequate coping strategies and increased stress. 

Two studies found parents with ASD children utilize maladaptive coping strategies to shield their children’s stress (i.e., coping for active avoidance and escape avoidance) more than parents with typical children, which is the underlying cause of crisis in the homes of ASD children [24,26]. According to Seymour, Wood, Giallo, and Jellett [40], significant levels of tiredness, which were linked to high maternal stress, were highly linked to maladaptive coping. Cappe, Pedoux, Poirier, Downes, and Nader-Grosbois [44] found that French parents placed more focus on problem coping strategies than emotion-centered methods and sought social assistance. Begum, Islam, and Rahman [41] demonstrated that parents with a higher level of education more frequently accepted responsibility, and Pisula and Kossakowska [42] found that the frequency of stress and greater burdens related to childcare experienced increased with the deterioration of the parents’ sense of coherence.

In several parents with ASD children, meaningfulness was negatively impacted by a positive re-evaluation [42] and the rise in care-related parental stress had a strong correlation with an increase in reported ASD symptoms over all four areas [43].

### 3.7. The Correlation between Parenting Stress and QoL of Parents with ASD Children

Two studies analyzed the significant negative correlation between parenting stress and QoL in most areas (physical health, psychological well-being, and social relations), and found that a high level of parenting stress was correlated with a lower QoL. Four studies found that parents with ASD children had high levels of parenting stress and lower QoL than parents with typical children.

Furthermore, there are two factors that are known to moderate the correlation between parenting stress and QoL. The first is social support [15] and the second is self-compassion [22]. Another study looked at the factors that influence QoL, including everyday activities, interactions between families and couples, psychological well-being, occupation, social activities and relationships, and activities and relationships with the ASD child [44]. One study discovered that a lower cognitive stigma is linked substantially with a greater quality of life and overall satisfaction [47].

### 3.8. The Relationship between Parenting Stress, Positive Reappraisal Coping, and QoL

Six studies investigated the association between parenting stress, coping strategies, and QoL [15,21,22,39,47,48]. The results were varied but generally indicated that a high degree of parenting stress could be related to inefficient coping strategies and a low level of QoL (see Table 3). Two studies analyzing the relationship between parents’ stress and QoL showed that, in most areas (physical health, psychological well-being, and social relations), high levels of stress in parents with ASD children were correlated with lower QoL levels than parents with typical children, which was observed by four studies [21,39,46,47]. 

Five studies reported an overall association between coping strategies and QoL [15,38,48]. Cappe et al. and Pisula et al. [38,48] found that parents used two emotional regulation strategies to face the situation (i.e., self-soothing and deep exhalation), while Cappe, Bolduc, Rougé, Saiag, and Delorme [38] found that parents who leaned on their feelings and had more unwanted contact with their ASD children struggled more to live their everyday lives. On the contrary, parents who commonly used problem-solving mechanisms or sought social help had stronger relationships with their children and were happier. Social support has been known to mitigate the link between stress and QoL, for example, by moderating factors such as receiving social assistance and avoiding stressful circumstances, and it was revealed that taking responsibility mediated the association between stress and QoL [15]. Cappe, Pedoux, Poirier, Downes, and Nader-Grosbois [44] found that problem-focused coping techniques and the search for social support seemed to be the more effective coping strategies for helping parents understand challenges in life and lead to better-adjusted parents who are less disturbed.

### 3.9. Factors That Influence Relationship between Parenting Stress, Positive Reappraisal Coping, and QoL

This study found the factors that influence QoL, including everyday activities, interactions between families and couples, psychological well-being, occupation, social activities and relationships, and activities and relationships with the ASD child [44]. Whether or not parents are aware of their children’s problems [38] and have compassion for themselves are other indications of parenting stress and QoL [22]. In terms of parent QoL, one study discovered that lower cognitive stigma is linked substantially with a greater QoL and overall satisfaction [47].

## 4. Discussion

This review sought to further explore the relationship between parenting stress, positive reappraisal coping, and the quality of life in parents with ASD children. Moreover, it also examined the role of coping strategies as mediators for this relationship. Overall, based on the studies in this review, there appeared to be a significant correlation between parenting stress, positive reappraisal coping, and quality of life.

Parenting stress is an outcome of a series of assessments of the parenting role and a psychological reaction that arises due to parental strain. Furthermore, Abidin [49], Craig et al. [50], and Deater-Deckard [51] argued that the tension between parents usually arises from stress related to childcare. Moreover, when stress is a result of parents feeling powerless, they may not be able to cope with stress while taking care of their children and might seek help for their children (i.e., through schools, health provider units, therapy services, learning management and care, and contact with their teachers). 

This study revealed the following factors had an important effect on parental stress: behavior problems and emotional impairments of the child, age, lack of respite care, several ASD children in the household, race/ethnicity, family communication, hopeful thinking, social support, life satisfaction, parenting self-efficacy, the children’s lability, negative impressions of parents, and daily positive affect [25,27,32,37].

The present analysis review found that parenting stress has a positive significant correlation with positive reappraisal coping. High levels of parenting stress were correlated with negative reappraisal coping and led to the lack of knowledge of coping strategies [26]. Parenting stress was correlated with different coping mechanisms used by parents to address their issues in meaningful ways [24,26,42]. Nevertheless, variations in how coping mechanisms were assessed rendered categorizing coping strategies difficult in many of the reviewed studies.

Research on coping strategies conducted by Begum, Islam, and Rahman [41], Dardas and Ahmad [15], and Cappe, Pedoux, Poirier, Downes, and Nader-Grosbois [44] all found that active coping, acceptance, positive reappraisal coping and development, elimination of competitive activities, planning, and problem-focused coping strategies were the most frequently reported coping strategies used by parents with ASD children.

Conversely, a high stress level in parents with ASD children was significantly connected to a lack of knowledge of coping strategies (i.e., active avoidance, escape avoidance, maladaptive/emotion-focused coping) [26,40]. The findings of this investigation were supported by the research of Hall and Graff [24], who found that parental weariness is linked to insufficient coping techniques, and research by Pisula and Kossakowska [42], who found that parents prefer to use escape avoidance coping because caring for ASD children raises stress levels and depression and also decreases positive mood levels. This was related to findings from a review by Cappe, Bolduc, Rougé, Saiag, and Delorme [38], in which they found that it was harder for parents of ASD children to live their regular lives, and that parents who relied on their emotions to manage had a more damaged relationship with their ASD children. Similarly, parents using coping skills or seeking social assistance for overcoming problems were better and happier with their children. In one study, Pisula and Kossakowska [42] found that positive reappraisal coping negatively correlated with meaningfulness, whereas Begum, Islam, and Rahman [41] found that positive reappraisal coping was frequently used by parents to change their daily lifestyles in order to motivate themselves towards new and evolving goals in life. 

This study found that the education level was a moderator variable in the relationship between parenting stress and coping strategy. education The level affected parents’ tendency towards “accepting responsibilities”. When taking charge of the issues in their lives, parents with a high level of education felt more self-control and could make progress in rehabilitation more easily, rather than focusing on the past and blaming external forces for their problems. 

It was observed that there was a significant negative correlation between parenting stress and QoL in parents with ASD children [21,39,46,47]. Social support and self-compassion were moderating factors in the correlation between parenting stress and QoL [15,31]. According to Hosseininejad et al. [52], the availability of social support and the capacity to speak with others are critical for stress prevention and treatment.

The present systematic review found that parenting stress correlated with coping strategies and QoL. High levels of parenting stress led to ineffective coping strategies and low QoL [15,21,22,39,47,48]. The correlation between parenting stress and QoL was mediated by the coping strategies chosen by parents with ASD children (i.e., accepting responsibility coping strategy) [15]. Positive reappraisal coping led to positive feelings in the parent and affected their relationship with their ASD children [46]. Future studies on this group’s coping strategies and quality of life should try to gain a deeper understanding of these issues. 

The strengths of this study were that every attempt was undertaken to include all relevant studies, include a comprehensive search strategy, strict inclusion and exclusion criteria, critical appraisal, and analytical strategies that included an in-depth subgroup analysis to identify heterogeneity among the articles. Secondly, all included studies used the same diagnostic guidelines that minimized the confounding effect while pooling the data for the analysis. The limitation of this study was our search criteria to only papers published in English and papers that employed a cross-sectional study design. Therefore, our findings are restricted by the absence of more studies, such as those using an experimental study design, to explain the chronological sequence of identified associations within variables. Future studies should also evaluate the association between parenting stress, coping strategy, and QoL using an experimental design as part of the inclusion criteria. 

## 5. Conclusions

We found a significant negative correlation between parenting stress, positive reappraisal coping, and QoL in parents with ASD children. High levels of parenting stress resulted in low levels of positive reappraisal coping and a lower level of QoL. Positive reappraisal coping played a role as a mediator variable in the correlation between parenting stress and QoL in parents with ASD children.

## Figures and Tables

**Figure 1 healthcare-10-00052-f001:**
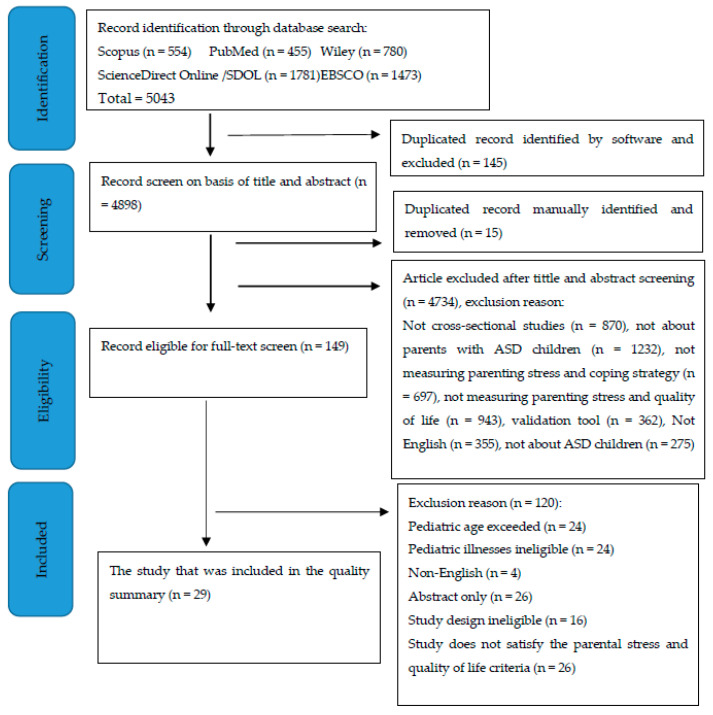
PRISMA flow diagram for the systematic review of parental stress, positive reappraisal coping, and quality of life of parents with ASD children.

**Table 1 healthcare-10-00052-t001:** Socio-demographic characteristics of the parents and children.

	Study	Study Design	Country	Participant	N (Sample Size) Male/Female	Age of Parents/Mean	Age of Children/Mean	Diagnosis
1	Ashworth et al. (2019)	Cross sectional study	England	Mother	265	Parents of children with WS (n = 107), DS (n = 79), and ASD (n = 79)	4–25 years	WS (n = 107); DS (n = 79); ASD (n = 79)
2	Costa et al. (2017)	Cross sectional study	Germany	Mother and father	78 ASD 37 (31/6); TD 41 (36/5)	ASD: 33–53 years (M = 41.29); TD: 26–49 years (M = 39.12)	ASD: 3–13 years (M = 9.27) TD: 4–13 years (M = 8.42)	ASD (n = 37) TD (n = 41)
3	Factor et al. (2017)	Cross sectional study	Virginia, US	Mother	27	34–51 years (M = 39.10)	7–12 (M = 8.88)	ASD
4	Hou (2018)	Cross sectional study	Taiwan	Mother	102	Mother with ASD children (M = 34.73); mother with DD children (M33.04)	ASD (M = 31.81 months), DD (M = 30.16 months)	ASD (n = 51) DD (n = 51)
5	Kim et al. (2020)	Cross sectional study	USA	Parents	1131 = Hispanic n = 125 (M = 25(20.5), F = 97(79.5)); White n = 790 (M = 208 (26.6), (F = 574(73.4.)); Black n = 80; (M = 15 (19.2), F = 63(80.8)); Asian n = 48 (M = 25 (54.3), F = 21(45.76); multi race n = 88 (M = 21(24.4), F = 65(75.6))	Hispanic (42.41), White 790(44.27), Black 80 (44.6), Asian 48 (44.30), multi race 88 (43.47)	Hispanic (M = 9.80), White (M = 11.61), Black (M = 10.64), Asian (M = 10.50), multi race (M = 10.64)	ASD
6	Lu et al. (2018)	Cross sectional study	China	Parents	479	36.59	3–18 (6.68)	ASD
7	Pisula and Kossakowska (2010)	Cross sectional study	Poland	Mother and father	N = 202: parents with ADS children (49) and parents with TD children (52)	26–47 years; parent with ASD children (Mother (39.56) Father (41.87)); parent with TD (mother (39.94), father (41.62)	5–17 ASD (10.24), TD (10.21)	ASD (n = 26), TD (n = 29)
8	Mohakud (2019)	Cross sectional study	India	Mother	Mother of ASD children = 23; mother of CP children = 21	36–65 years	ASD (M = 4.6), CP (M = 3.7)	ASD (n = 23), CP (n = 21)
9	Pattini et al. (2019)	Cross sectional study	Italy	Mothers	N = 39: mother of TD children (n = 15); mother of ASD children (n = 15)	Mother of ASD children (M = 39.5); mother of TD children (M = 38.3)	3–11 years; TD children n = 15(5.7), ASD children n = 15 (39.5)	ASD (n = 15), TD (n = 15)
10	Gong (2015)	Cross-sectional study	China	Parents	196	Fathers: 19–62 years (M = 37.08); Mothers: 25–50 years (M = 34.26)	23–144 months (M = 77.73 months)	ASD
11	Rivard et al. (2014)	Cross sectional study	Canada	Mother and father	N = 236 (118 mothers and 118 fathers)	Not mentioned	2.9–5 years-old (SD = 0.61)	ASD, Asperger’s syndrome, Pervasive Developmental Disorders-Non-Specified (PDD-NOS)
12	Rodriguez et al. (2019)	Cross sectional study	USA	Mother and father	187	Parents had an average age of 39.68 years; mothers (M = 38.7) and fathers (M = 40.7)	5–12 years	ASD, intellectual disability
13	Siu et al. (2019)	Cross sectional study	Hongkong	Parents	N = 731 (177 parents of ASD children and 554 parent of TD children’s children)	41.5 years old	6–11 years (M = 8.4 years old for ASD, M = 8.6 years old for TD)	ASD (n = 177), TD (554).
14	Tomeny (2017)	Cross sectional study	USA	Mother/female caregivers	111	31–60 years (M = 44.1)	3–17 years (M = 11.98)	ASD (56% with ASD, 21% with Asperger’s Disorder, 23% with pervasive developmental disorder, not otherwise specified (PDD-NOS))
15	Begum et al. (2020)	Cross sectional study	Bangladesh	Parent	44	Not mentioned	Not mentioned	ASD
16	Lai et al. (2015)	Cross sectional study	India	Parents	N = 136 (73 parent of ASD children, 63 parent of TD children)	43.68 years; ASD n = 43 (46.10); AS n = 15 (46.00); PDD-NOS n = 15 (48.30); TD n = 63 (43.68)	ASD n = 43 (14.10); AS n = 15 (12.90); PDD-NOS n = 15 (13.25); TD n = 63 (10.80)	ASD, AS, PDD-NOS, TD
17	Cappe et al. (2020)	Cross sectional study	Canada	Parent	87	28–56 years (M = 41.25)	4.46	ASD
18	Cappe et al. (2017)	Cross sectional study	Canada	Parent	77	29–56 years (M = 40.49)	M = 9.62 years old (under 6 years 14.3%; 6–9 years 27.3%; 9–12 33.8%; over 12 years 24.7%)	ASD
19	Chu et al. (2020)	Cross sectional study	Malaysia	Parent	110	31–40 years	2–18 years	ASD
20	Hsiao et al. (2017)	Cross sectional study	USA	Parents	236	Not mentioned	<5 years (n = 29) 5–12 years (n = 150) 13–18 years (n = 63) >18 years (n = 24)	ASD
21	Nuske et al. (2018)	Cross sectional study	USA	Parents	71 = (ASD n = 43, TD n = 28))	Parent of ASD children = 40.89 years; parent of TD = 41.79	ASD (24–59 months), TD (24–61 months)	ASD (n = 43), TD (n = 28)
22	Pisula and Dorman (2017)	Cross sectional study	Poland	Mothers and fathers	202 (49 mother–father dyads; ASD), (parents of TD 52 mother–father dyads)	ASD parents = mother (39,56 years), father (41.87 years); TD parents = mother (39.94 years), father (41.62 years)	ASD children (10.24); TD children (10.21)	ASD (n = 49), TD (n = 52)
23	Thullen and Bonsall (2017)	Cross sectional study	Columbia	Parent	113	29–64 years	5–13 years	ASD
24	Hall and Graff (2012)	Cross sectional study	USA	Mother and father	70 Mothers (n = 48) and father (n = 22)	40 years	3–21 years	ASD
25	Ishtiaq et al. (2020)	Cross sectional study	Pakistan	Mother and father	300: n = 200 parents of HI and 100 parents of ASD children	20–60 years (40.56)	Not mentioned	HI (n = 200) ASD (n = 100)
26	Seymour et al. (2013)	Cross sectional study	USA	Mothers	65	36.09 years	2–5 years	ASD
27	Shepherd et al. (2018)	Cross sectional study	New Zealand	Father and mother	178 fathers (n = 19) and mothers (n = 159)	29–60 or over years (M = 45.27)	Up to 9 years: n = 56 (35%); 10–19: n = 78 (49%); 20–29: n = 17 (11%); 30 or over: n = 7 (4%)	ASD
28	Bohadana et al. (2019)	Cross sectional study	Australia	Father and mother	139 Fathers (n = 19) and mothers (n = 120)	Mother (M = 39.49); father (M = 35.16)	6–12 years	ASD
29	Dardas and Ahmad (2015)	Cross sectional study	Jordan	Mother	184	21–69 years (M = 37)	Under 12 years	ASD

ASD—autism spectrum disorder; TD—typically developing; HI—hearing impairment; DD—developmental delay; AS—Asperger’s syndrome; PDD-NOS—pervasive developmental disorder not otherwise specified.

**Table 2 healthcare-10-00052-t002:** Self-administered measurement tools used to examine parenting stress, positive reappraisal coping, and QoL status.

Measures	Description of the Tool	Reviewed Studies Using This Tool
**Parenting Stress Instrument**		
The Parenting Stress Index-Short Form (PSI-SF)	Used to evaluate a degree of parenting stress based on Abidin (1995). Consist of 12 items on a 5-point Likert scale from ‘strongly agree’ (1) to ‘strongly disagree’ (5).	Siu et al. (2019), Pattini et al. (2019), Lu et al. (2018), Hou, (2018), Gong, (2015), Bohadana et al. (2019), Rivard et al. (2014), Lai et al. (2015), Thullen and Bonsall, (2017), Hall and Graff, (2012), Dardas and Ahmad, (2015)
The Genetic Syndromes Stressors Scale (GSSS)	Used to measure parental stressors relating to rare genetic disorders. Consist of 14 items on a Likert scale ranging from 0 (not at all stressful) to 3 (extremely stressful).	Ashworth et al. (2019)
Family Resilience The 2016 NSCH	Used to measure the level of parenting stress and family resilience. Consist of 3 items with 4 response categories: (1) never, (2) rarely, (3) sometimes, and (4) usually or always. A higher score indicated higher parenting stress and higher levels of family resilience.	Kim et al. (2020)
The Burden Interview	Used to measure the level of parenting stress. Consists of 29 items, rated on a four-point scale from 0 (not at all) to 3 (extremely). A higher score indicating greater levels of parenting stress.	Rodriguez et al. (2019)
The Parental Stress Scale (PSS)	Used to assess the level of parenting stress caused by the parental task, circumstances, and difficulties. Consists of 18-item questionnaire.	Hsiao et al. (2017), Istihaq et al. (2020)
Perceived Stress Scale (PRSS)	Used to measure an individual’s perception of their intensity of response for several daily stressful situations. Consists of 23 self-report questions. Each question contains a statement that describes a potentially stressful situation and three choices describing potential responses.	Mohakud, (2019); Factor et al. (2017)
The Questionnaire on Resources and Stress–Short Form (QRS-F)	Used to assess self-report parental adaptation and coping related to raising a child with developmental delays, physical handicaps, or chronic illness. Consists of 52 items.	Tomeny, (2017)
Heart Rate Variability (HRV) Index	Used to measure emotional responses throughout the autonomous nervous system. High resting HRV is thought to indicate readiness to respond to environmental demands and emotion regulation capacity. The high-frequency (HF, 0.15–0.40 Hz) component of these changes was designated as the most reliable indicator of HRV and lower values reflect less emotional regulation.	Costa et al. (2017)
The Appraisal of Life Events Scale (ALES)	Used to assess the dimension of perceived stress (threat, loss, and challenge. Consisted of 16 items that rated on a 5-point Likert scale, ranging from 0 (“not at all”) to 5 (“Extremely”). The higher the score, the more parents perceive their situation as a threat, a loss, or a challenge.	Cappe et al. (2017)
The short form of the Questionnaire of Resources and Stress for Families with Chronically Ill or Handicapped Members (QRS–S)	Used to measure three main areas of stress: child problems (18 items), personal problems (30 items), and family problems (18 items). For each of the 66 items, participants circle one of two responses: true/false. Higher scores indicate higher stress.	Pisula, and Dörsmann, (2017)
The depression, anxiety, and stress scale-21 (DASS-21)	Used to assess the negative emotional states of depression, anxiety, and stress. Items rated on a 4-point scale ranging from 0 = “did not apply to me at all” to 3 = “applied to me very much, or most of the time”.	Seymour et al. (2013), Lai et al. (2015)
**Positive Reappraisal Coping Instrument**		
A revised version of a coping checklist	Used to measure the coping strategies. Consists of eight groups: confrontive coping, distancing, self-controlling, seeking social support, accepting responsibility, escape avoidance, planful problem-solving, and positive reappraisal. For each group, participants were asked to rate the strategy using a four-point scale ranging from zero to three (0–3) (“do not use” to “use frequently”)	Begum et al. (2020)
The Brief Coping Orientation of Problems Experienced (Brief-COPE)	Used to assess 14 differences coping strategies used by parents. Consists of 28 items. Items are rated on a four-point rating scale (i.e., from 1 = “I have not been doing this at all” to 4 = “I have been doing this a lot”.	Lai et al. (2015), Seymour et al. (2013); Shepherd et al. (2018); Dardas and Ahmad, (2015)
Ways of Coping Questionnaire (Folkman and Lazarus 1988)	Used to measure coping strategies used by parents. Consists of 66 items. The items grouped into eight scales: confrontive coping, distancing, self-controlling, seeking social support, accepting responsibility, escape avoidance, planful problem-solving, and positive reappraisal.	Pisula and Kossakowska, (2010)
The French version of the Ways of Coping Checklist (WCC-R)	Used to evaluate coping strategies of parents with PDD children. Consists of 27 items rated on a four-point scale (0 means “no” and 3 means “yes”). Scores ranged from 0 to 21. A higher score indicated that the respondent coped with stressful events by seeking assistance, information, advice, sympathy, or emotional support from others.	Cappe et al. (2017); Cappe et al. (2020)
A short version of the Coping Strategies Inventory (CSI-SF)	Used to assess the level of stress and coping strategies used by parents with HI and ASD children. Consists of 16 items and 4 subscales: problem-focused engagement (PFE); problem-focused disengagement (PFD); emotion-focused engagement (EFE); emotion-focused disengagement (EFD).	Ishtiaq et al. (2020)
**Quality of Life (QoL) Instrument**		
World Health Organization Quality of Life Assessment Questionnaire (WHOQOLBREF)	Used to measure the level of quality of life of parent. Latent variable composed of four subscales (physical, environmental, psychological, and social relationships). Consists of 26 items. The higher score indicates better quality of life.	Pisula and Dörsmann, (2017)
The Family Quality of Life Scale (FQOL)	Used to assess parents’ perceived satisfaction with their quality of life. Consists of 25-item scale with 5 sub-scales (family interaction, emotional well-being, parenting, physical/material well-being, and disability-related support). The items are rated on a 5-point scale ranging from 1 (very dissatisfied) to 5 (very satisfied). A higher total score refers to a higher family QoL.	Hsiao et al. (2017)
The CarerQol-7D scale	Used to assess a comprehensive description of the caregiving situation and parent’s well-being. Consists of five negative and two positive dimensions. The five negative dimensions include (i) relational problems, (ii) mental health problems, (iii) problems combining daily activities with care, (iv) financial problems, and (v) physical health problems. The two positive dimensions are (i) fulfillment from caregiving and (ii) support with lending care.	Chu et al. (2020)
The Quality of Life in ASD Scale	Used to measure the parental perception of quality-of-life parents of children with ASD. Consists of 28 items on a 5-point Likert scale ranging from 1 (not very much) to 5 (very much). Higher scores indicate the better parent-reported quality of life.	Bohadana et al. (2019)
The Pediatric Quality of Life (PedsQL) Inventory	Used to assess quality of life of parents in 4 domains: physical, social, emotional, and cognitive functioning. Consists of 20 items. Item is scored on a 5-point scale, ranging from 0 = never a problem to 4 = almost always a problem. Items are reversed scored and linearly transformed to a 0–100 scale as follows: 0 = 100; 1 = 75; 2 = 50; 3 = 25; 4 = 0. Higher scores indicate higher parent quality or life and family functioning.	Nuske et al. (2018)

HRV—heart rate variability; HI—hearing impairment.

**Table 3 healthcare-10-00052-t003:** Interest variable, analytic technique, and key findings of the studies included.

Author, Year	Variables of Interest	The General Approach to Analysis	Main Finding
Ashworth et al. (2019)	Parenting stress: rare genetic disorders parental stresses	One-way ANOVAs and Bonferroni post hoc analyses	The stress levels of parents with WS, DS, and ASD were similar. (F (2, 262) =0.278, *p* = 0.757, η^2^ = 0.002). The circumstances influencing this stress varied among groups.
Costa et al. (2017)	The stress of parents: individual differences in HRV + capability for emotion control Well-being of parents: many facets of emotional problems	Independent sample *t*-tests	Parents of children with ASD reported worse subjective well-being (higher DERS score) (t (64) = 2.36, *p* 0.05, r = 0.28), as well as more physiological stress (lower resting HRV) (t (45) = 2.55, *p* 0.05, r = 0.36). Parents’ assessments of their children’s lability/negativity contributed to the physiological stress of parents of children with ASD (t (76) = 5.64, *p* 0.001, r = 0.54).
Factor et al. (2017)	Parenting Stress: self-reports of stress (the increased support for stress in self-reports would indicate higher stress)	Pearson correlations	There was no link between ASD characteristics and self-reported stress (=0.03, *p* = 0.35). There was a significant positive main impact for ASD features on HRV (=0.643, *p* = 0.01), which means that children with more ASD traits had bigger increases in mother’s HRV.
Hou (2018)	Parenting stress: stress for parents having children under the age of 12	Pearson’s correlation analyses	Parental stress and depressive symptoms were shown to be higher in women with ASD children 281.76 (36.03) than in mothers with DD 281.76 (36.03). Mothers of children with ASD at the age of 32 months had symptoms that were similar to mild depressive symptoms (ASD = 13.98 (8.70) DD = 10.35) (9.16).
Kim et al. (2020)	Parenting stress: three parenting stress items Family resilience: four family resilience items	ANOVA or chi-square tests	Family resilience was adversely associated with parenting stress (=0.17, *p* = 0.01). Higher parenting stress was linked to more child behavioral difficulties (=0.28, *p =* 0.01). When compared to parents who stated they received emotional support, those who stated they did not, had higher levels of parenting stress (=0.31, *p =* 0.01). There was negative relationship between family resilience and parenting stress for parents of white children (β = −0.14, *p* < 0.05) and African American children (β = −0.38, *p* < 0.05), although the relationship was stronger for parents of African American children
Lu et al. (2018)	Parent stress: three factors (parenting difficulty, relationship between parent and child, and problematic children and parental stress)	Spearman correlation test	Life satisfaction of parents of children with ASD was significantly negatively correlated with parenting stress (r = 0.391, *p* = 0.01) and significantly positively correlated with social support (r = 0.372, *p =* 0.01), whereas parenting stress was significantly negatively correlated with social support (r = 0.322, *p =* 0.01). Parenting stress predicted life satisfaction (β = −0.391, *p* < 0.01), parenting stress predicted social support (β = −0.322, *p* < 0.01), parenting stress and social support predicted life satisfaction (β = −0.302, *p* < 0.01; β = 0.275, *p* < 0.01). Parenting stress partially mediated the relationship between social support and life satisfaction.
Mohakud (2019)	Parenting Stress: perceived stress in mothers	Mann–Whitney U Test	Mothers of children with ASD showed greater levels of parental stress and anxiety than mothers of children with cerebral palsy (ASD: 30.217.34, CP: 24.767.34). Parental stress was positively associated with child problem behavior (*p =* 0.05, r = 0.353).
Gong (2015)	Parenting stress: assessing stress in parent–child relationships (both child and parent)	*t*-tests	The strongest predictors of parenting stress were the age of the kid, behavioral issues, and maternal anxiety symptoms, which explained 54.9 percent of the variations. PSI = 188.765 + 0.630 × 1 + 1.249 × 2 + 0.231 × 3 was the multiple linear regression equation. Parents of ASD children reported higher total stress levels than parents of typically developing children (t = 13.76, *p* = 0.000). Mothers of ASD children had higher SDS scores than ASD fathers, who in turn had higher SDS scores than control parents (F = 17.561, *p* = 0.000). Mothers of autistic children scored higher on the SAS than fathers of autistic children, who scored higher on the SAS than control parents (F = 17.535, P = 0.000). In mothers of TD children, state anxiety levels were considerably lower at the end of the PST compared to pre-PST levels (t = 3.34, *p* = 0.05), but not in mothers of ASD children.
Pattini et al. (2019)	Stress: disorder related to the role of parenting coping strategies: several adaptive coping strategies	Shapiro–Wilk Test	Mother of ASD children reported higher levels of state anxiety than mother of TD, both prior to (mother of ASD = 41.3 ± 1.9 vs. mother of TD = 32.8 ± 0.8, t = 4.05, *p* < 0.05, d = 1.51) and after (mother of ASD = 39.5 1.8 vs. mother of TD = −28.5 1.0, t = 5.34, *p =* 0.05, d = 1.95) the PST, mothers of ASD children reported higher levels of state anxiety than mothers of TD. During a psychosocial stress test, mothers of children with ASD (mothers of ASD, n = 15) and mothers of typically developing children (n = 15) were assessed for parental stress levels, psychological characteristics, and coping strategies, as well as measures of heart rate, heart rate variability, and cortisol. Mothers of ASD reported higher levels of state anxiety than mothers of TD both before (mothers of ASD = 41.3 1.9 vs. mothers of TD = 32.8 0.8, t = 4.05, *p =* 0.05, d = 1.51) and after (mothers of ASD = 41.3 1.9 vs. mothers of TD = 32.8 0.8, t = 4.05, *p =* 0.05, d = 1.51). In terms of HR values, there was a significant group effect (F = 5.61, *p =* 0.05), with mothers of ASD having considerably higher mean HR values than mothers of TD.
Rivard et al. (2014)	Parenting stress: the perceived stress of fathers and mothers with ASD children	*t*-tests and bivariate analyses, regression analyses	Fathers reported being more stressed than mothers. t (117) = −3.83, *p* = 0.01 (M of 118.35 and 112.38, respectively). The stress levels of parents were found to be connected to the child’s age, academic achievement, the severity of autistic symptoms, and behavioral modification. The severity of autistic symptoms and the child’s sex predicted father stress, but not mother stress.
Rodriguez et al. (2019)	Parenting stress: parental and childcare-related personal distress and difficulties Child behavior issues: children with ASD-level behavioral problems	Paired sample *t*-tests	A significant difference between mother and father reports of parenting stress at T1 (t = 3.16, *p* < 0.01), T2 (t = 2.88, *p* = 0.01), and T3 (t = 3.15, *p* < 0.01), and T4 (t = 2.23, *p* = 0.03). Parenting stress was adequately stable for mothers and fathers across time points (mother T1–T2: β = 0.82, *p* < 0.001; T2–T3: β = 0.50, *p* < 0.001; T3–T4: β = 0.54, *p* < 0.001; father T1–T2: β = 0.72, *p* < 0.001; T2–T3: β = 0.66, *p* < 0.001; T3–T4: β = 0.58, *p* < 0.001).
Siu et al. (2019)	Parenting stress: the stress level reported by the parent directly related to parenting based on personal circumstances	Bivariate correlations	Significant indirect effects of autistic symptoms and internalizing difficulties on parenting stress (β = 0.13; SE = 0.07; = 0.09, *p* = 0.05), externalizing problems (b = 0.17; SE = 0.07; = 0.12, = 0.05), and prosocial behaviors (b = 0.18; SE = 0.03; = 0.12, *p* = 0.001). Parents with ASD children reported higher stress levels than parents with TD children. It could be linked to the observation that children with TD are significantly less prosocial than children with ASD and are free of problem habits in all respects (a satisfactory goodness of fit, 2 (115) = 321.63, *p* = 0.001; CFI = 0.96; TLI = 0.95; RMSEA = 0.05; SRMR = 0.04).
Tomeny (2017)	Parenting stress: parental adjustment and coping associated with raising children with developmental delays, physical disabilities, or chronic diseases	Zero-order correlation analyses	The intensity of ASD symptoms was associated with maternal parenting stress (r = 0.47, *p* = 0.001) and maternal psychopathology symptoms (r = 0.29, *p* = 0.002). There was a moderate relationship between maternal parenting stress and maternal psychopathology symptoms (r = 0.37, *p =* 0.001). ASD symptoms had an indirect effect on mother psychopathology symptoms via maternal parenting stress, with a point estimate of 0.07 (95 percent confidence interval = (0.02, 0.14)).
Begum et al. (2020)	Coping: establish coping strategies based on problems and emotions	Chi-square test	All parents choose their abilities in distinct types of techniques. There were no significant relationships between education and confrontive coping (*p* = 0.529), distancing coping (*p* = 0.258), self-controlling coping (*p* = 0.547), seeking social support (*p* = 0.353), escape avoidance tendency (*p* = 0.541), planful problem solving (*p* = 0.514), or positive reappraisal (*p* = 0.158) according to the Pearson chi-square analysis. There was a statistically significant link between education and taking on responsibilities (*p* = 0.02). The majority of parents were “at least” attempting to resolve their problem (32/44), while a tiny percentage were upset (8/44). This finding suggested that most parents had a positive attitude about their child’s predicament and were seeking additional and novel medication and therapies to enhance their child’s management.
Lai et al. (2015)	Coping: the use of broad-based dysfunctional coping methods and adaptive coping Parent stress: stress-related actions and feelings based on a parent–child interaction (P–CDI) scale and a problematic child measure based on three sub-scales of a child’s parental distress (DC) scale	MANOVA and chi-square analyses	Parents of children with ASD reported much higher levels of parenting stress, despair, and maladaptive coping than parents of typically developing children. F (30, 362) = 2.47; gp2 = 0.17. Wilks’ Lambda = 0.58; F (30, 362) = 2.47; gp2 = 0.17. The error variance was estimated to be 17 percent.
Pisula and Kossakowska (2010)	Coping: coping strategies	Correlation analysis	Parents with ASD children utilized escape-avoidance to protect themselves from stress at a higher rate than their parents (F (1,104) = 4.69; *p* = 0.033; n partial = 0.04). Parents of children with ASD had a lower overall SOC than parents of children who were usually developing. SOC had no gender differences (F (1,104) = 10.459, *p* = 0.002; n^2^ partial = 0.09). Parents of children with ASD had lower meaningfulness scores than parents of typically developing children (F (1,104) = 8.81; *p* = 0.004; n^2^ partial = 0.08).
Hall and Graff (2010)	Parenting Stress: parents react to (1) parent stress, (2) interactive parent–child dysfunction, and (3) kid difficulty in areas where parents are affected Coping: coping health inventory for parents	Pearson product-moment correlations, regression analysis	There was an association between increased maladaptive behaviors of children and parental stress (positive relationships existed between the Maladaptive Behavior Index and PSI-SF (r = 0.464, *p* = 0.000), and between the Internalizing Maladaptive Behavior subscale and PSI-SF (r = 0.547, *p* = 0.000). The Internalizing and the Externalizing Maladaptive Behavior subscales were positively associated (r = 0.422, *p* = 0.000)). Parents could employ tactics to control behaviors, but they could not utilize or know procedures to govern comportments. Parental stress was the value of the conditions attributable to the family. Circumstances included additional stressors (maladaptive behaviors), which were regarded to be the cause of an ASD-diagnosed family crisis.
Seymour et al. (2013)	Stress: the negative emotional states of stress over the past week Coping: frequently parent engage in each of the behaviors and cognitions when coping with a specific stressful situation	Chi-square test	There was no significant relationship between child behavior problems and maladaptive coping. Maladaptive coping was associated with higher maternal stress x^2^ (1, N = 65) = 0.68; *p* = 0.410; GFI = 1.00; AGFI = 0.95; CFI = 1.00; TLI = 1.04; RMSEA =0.00 (0.00–0.31). The model significantly accounted for 43% of the variance in maternal stress (R2 = 0.43, *p* < 0.001). Problems with child behavior may lead to parental exhaustion and may affect the use of inadequate coping strategies and increased stress (β = 0.32, t = 2.68, *p* = 0.007). High levels of fatigue were found to be strongly linked to inefficient coping mechanisms, which were linked to high levels of mother stress. Maternal fatigue totally moderated the association between child behavior problems and maternal stress (β = 0.08, t = 0.83, *p* = 0.41).
Shepherd et al. (2018)	Parent stress: frequently experienced by parents with ASD children Coping strategies: 14 different coping styles	Preliminary correlational analyses	There were high correlations (r > 0.5) between ASD care-related stress ratings and core ASD symptoms, with an increase in the perceived severity of ASD symptoms across all four categories being linked to an increase in the parents’ stress perception. ASD care-related stress ratings and core ASD symptoms had high associations (r > 0.5) with an increase in the perceived severity of ASD symptoms across all four categories associated with an increase in the parents’ stress perception. There were clear relationships between ASD stress and core autistic symptoms, with higher perceptions of ASD symptoms being linked to more parenting in all four categories. The link between restricted/ritualized conduct symptoms and stress was reduced thanks to coping measures. Strong (r > 0.5) positive correlations existed between the four AIM symptom measures and the ASD care-related stress scale, but only minor (r = 0.03) positive correlations existed between the AIM and the CRA measures.
Cappe et al. (2020)	Coping strategies: the different coping strategies by parents with ASD children Quality of Life: there has been considerable correlations between ASD stress evaluations and key ASD symptoms in all four categories with increasing perceptive of ASD symptoms Perceived stress: the factors that influence stress in their daily lives	Pearson’s chi-squared tests	Parents generally used more problem-focused coping strategies (M = 59.9; SD = 20.91) than emotion-centered strategies (M = 45.5; SD = 21.22) or searching for social support (M = 61.5; SD = 24.56). French parents, however, made significantly less use of available social support (M = 52.71; SD = 24.74; F (2, 84) = 5.87, *p* < 0.05). The impact of ASD on parents’ quality of life appear4r to be greater in French parents. (M = 54.04; SD = 14.26). However, the three groups did not differ significantly. In daily activities, the parent’s life quality was increasingly influenced (M = 65; SD = 18.45), followed by family and couple relationships and activities (M = 54.13; SD = 17.22), psychological well-being (M = 51.75; SD = 18.73), social activities and relationships (M = 51.54; SD = 21.45), professional activities and relationships (M = 47.88; SD = 28.85), activities and relationships with the child with ASD (M = 39.3; SD = 20.35), and personal fulfilment (M = 33.42; SD = 21).
Ishtiaq et al. (2020)	Parenting Stress: a level of stress using self-reporting of stress Coping: coping strategy used by parent	Descriptive analysis	Parents of ASD children had higher parenting stress and coping strategies than parents of HI children (ASD = 48.92 + 11.22; HI = 47.4412.85). Parents of high-functioning children utilized problem-focused disengagement (24.25), while parents of autistic children used the most common method (27.4), followed by emotion-focused strategy.
Chu et al. (2020)	Quality of Life: a detailed care situation description with a well-being informal care assessment	Pearson correlation	A higher level of affective stigma was linked to more stress (*p =* 0.001) and poor physical health (*p =* 0.05). Stress was found to have a moderate negative correlation of −0.62 and −0.57 with both quality of life and overall happiness, while quality of life had a positive, moderate link with overall happiness (r = 0.46). A lower degree of cognitive stigma was associated with improved overall happiness and quality of life (both *p* 0.05).
Hsiao et al. (2017)	Parenting Stress: positive components (examples of emotionality, personal enrichment, growth, and negative parenting components) Quality of Life: the degree in which parents are satisfied with their relationship with service providers	Structural equation modelling (SEM)	Parenting stress had a negative relationship with family quality of life (r = 0.494) and each of the five-family quality of life dimensions. The relationship between parental stress and family–teacher partnerships was not significant (r = 0.093). The parental satisfaction concerning family quality of life had an effect on the perceived parental stress level (β = −0.51, *p* < 0.05). The perceived parental stress level had a direct effect on the parental satisfaction concerning family quality of life (β = −0.46, *p* < 0.05). From the results of the second model, the perceived parental satisfaction concerning family quality of life had a direct effect on the perceived parental stress level (β = −0.48, *p* < 0.05).
Nuske et al. (2018)	Quality of life: measure of QoL for parent of children with clinical symptoms	Regression models	When compared to TD children, children with ASD demonstrated considerably higher levels of externalizing, but not internalizing, behaviors (β − 0.47. *p* < 0.001 ***). Parents of children with ASD had significantly lower quality of life (β − 0.60, *p* < 0.001 ***) and family functioning scores (β − 0.56, *p* < 0.001 ***) compared to those of TD children.
Pisula and Dorman (2017)	Parenting Stress: three key stress areas: problems with children, personal concerns, and family issues The Quality of Life: subjective evaluations of QoL in four areas: physical, psychological, social, and environmental health	A correction for multiple comparisons	The parents with ASD children were more stressed (27.17 (9.78)) than parents of children typically developing (15.56 (6.49)). Mothers with ASD children had higher stresses (29.94 (10.1)) than the fathers (24.41 (8.71)), whereas there were no variations in stress between mothers (15.56 (6.49)) and fathers (15.34 (6.04)) with typical children. A parent of an autistic child had a lower quality of life than a parent of a typically developing child in the majority of the dimensions studied (physical health F (1.201) = 9.849; *p* < 0.01; η^2^ = 0.047; psychological health F (1.201) = 213.559; *p* < 0.001; η = 0.064; social relationships domain F (1.198) = 15.806; *p* < 0.001; η^2^ = 0.074). There was a link between family functioning, parenting stress, and quality of life. The more negatively parents rated family functioning, the more parenting stress they experienced (β = 0.168, t = 2.341, *p* = 0.05) and the lower the QoL (β = 0.36, t = 4.622, *p* < 0.001). The level of overall stress was linked to family functioning (β = 0.168, t = 2.341, *p* < 0.05), as well as with family (β = 0.18, t = 2.259, *p* < 0.05) and personal parenting stress (β = 0.171, t = 2.4, *p* < 0.05).
Thullen and Bonsall (2017)	Parenting Stress: three domains of PS ((1) parent stress (2) dysfunction between parents and children, and (3) difficulties in children, parents responded to items) The Quality of Life: the quality of family parenthood	Pearson correlation	There was correlation between co-parenting and parenting stress. (0.29 ** *p* < 0.01) Disruptive mealtime behavior correlated with parenting stress (0.49, *p* < 0.01) and parents’ quality of life (0.04, *p* < 0.05).
Bohadana et al. (2019)	Parenting stress: parental dysfunction, parental dissatisfaction, and difficult children as three levels of parental stress The Quality of Life: perspective of QoL parent	Two multiple regression analyses	The significant double ABCX model predictor variables associated with parenting stress were child ASD symptoms, self-injurious behavior severity, aggressive behavior frequency and severity, stereotyped behavior frequency, pile up of demands, social support, active avoidance coping, parental self-efficacy, and parental perceptions. These predicted a significant 64.3% of the variance in parenting stress (F change (10, 125) = 22.49, *p* < 0.001). Parental stress was substantially predicted by perceptions (sr^2^ = 0.23), social support (sr^2^ = 0.03), and parental self-efficacy (sr^2^ = 0.02). Parental perceptions, social support, and the negative self-compassion component were the only significant predictors of parenting stress (F change (2123) = 27.39, *p* = 0.001). In both positive (B = 0.25. t = 2.71, 0.008, *p* = 0.01) and negative (B = −11. t = −1.16, 0.248, *p* = 0.01) dimensions, self-compassion added a significant amount of variation to the prediction of the quality of life.
Dardas and Ahmad (2015)	Parenting stress: stress in parents with ASD children under 12 years of age Coping: eight measurements in the subscale QoL: quality of life in general Quality of Life: an impression of their QoL	Bivariate and multivariable regression	Two types of coping methods were shown to be significantly predicted by stress: escape avoidance (t = 6.40; *p* = 0.001) and accepting responsibility (t = 4.55; *p* = 0.001). Accepting responsibility (−3.271; *p* = 0.001) and avoiding escape (t = −5.314; *p* = 0.001) had significant negative correlations with QoL. The association of stress and QoL reduced significantly when ‘accepting responsibility’ was added to the hierarchical equation. The coping strategy ‘escape avoidance’ in the second step of regression remained a significant predictor; thus, it was not considered as a mediator (−0.739; *p* = 0.461).

Ws: Williams syndrome; DS:down syndrome; ASD: autism spectrum disorders; TD: typically developing; HRV:heart rate variability; DD:developmental delay; ** *p* < 0.01; *** *p* < 0.001.

## Data Availability

The data presented in this study are available on request from the corresponding authors.

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
