# Peer review of "The Association between Parenting Stress, Positive Reappraisal Coping, and Quality of Life in Parents with Autism Spectrum Disorder (ASD) Children: A Systematic Review"

_healthcare, 2021, doi:10.3390/healthcare10010052_

Round 1
Reviewer 1 Report
Dear authors/editor, here are some minor comments:
- It is recommended to use the terminology of the DSM5, which refers to autism spectrum disorders (ASD), much better than Autism.
- The objectives of your paper should close the introduction, I am sure you can write them in a clearer way.
- Perhaps the section that the authors name as 2.2. Criteria for Inclusion and Exclusion would be better represented by 2 Eligibility criteria.
- I think that after describing the review process, where the number of studies is indicated, the PRISMA figure should be placed (before the table).
- Generate the list of references in the style recommended by the journal.
Author Response
Response to Reviewer 1 Comments
The association between parenting stress, positive reappraisal coping, and quality of life in parents with Autism Spectrum Disorder (ASD) children: An Analysis Review
Dear reviewer,
First of all, we would like to thank you for your constructive inputs for our manuscript. It has helped us to improve the manuscript, and we clarify text that might be confusing for the reader. We tried to revise the text according to reviewers suggestion. Hopefully, it will be answering the points raised by you. In the response for the reviewer comment we repeat your comments, which point by point. Your comments are followed by our response in red color and the direction associated with changes made in the text. You will find that we adapted the text in all cases. We hope that you will appreciate the new version and hope to hear from you soon.
Thank you for your consideration of this manuscript. I look forward to hearing from you.
Sincerely,
Assoc. Prof. Mein-Woei Suen, Ph.D.
Department of Psychology, College of Medical and Health Science, Asia University, Taiwan blake5477@yahoo.com.tw or blake@asia.edu.tw.
Point 1: First, It is recommended to use the terminology of the DSM5, which refers to autism spectrum disorders (ASD), much better than Autism.
Response 1: We thank the reviewer to point out this issue, we changed autism with ASD in all the text.
Point 2: The objectives of your paper should close the introduction, I am sure you can write them in a clearer way.
Response 2: Thanks for your suggestion, we have revised it. Please, see introduction with the yellow marks.
Point 3: Perhaps the section that the authors name as 2.2. Criteria for Inclusion and Exclusion would be better represented by 2 Eligibility criteria.
Response 3: Thanks for your suggestion, we have change criteria for inclusion and exclusion with eligibility criteria.
Point 4: I think that after describing the review process, where the number of studies is indicated, the PRISMA figure should be placed (before the table).
Response 4: We thank the reviewer to point out this issue. I have placed the PRISMA figure before the table.
Point 5: Generate the list of references in the style recommended by the journal.
Response 5: Thanks for your constructive suggestion. We have restructured references in the style recommended by the journal.
Reviewer 2 Report
This ms presents a review of the literature over the last 10 years on parental stress, coping strategies, and quality of life. The ms is generally well written and the methods employed in the review process appear to meet PRISMA Standards. Figure 1 is appropriate and provides adequate information regarding the literature search process. Use of Kappa for interobserver agreement was also an appropriate method.
Overall, my recommendation is to publish the paper with minor revision. Those revisions include:
“Children with autism” is the preferred term for this population. An alternative, not using person-first language would be “autistic children.”
Is it “ineffective use of coping strategies” or lack of knowledge of coping strategies and/or lack of resources to use coping strategies? This would be an appropriate addition to the discussion and interpretation of the results.
Table 3 does not seem to be necessary. Since all studies included all criteria (i.e., 100%), a note in the results section seems sufficient.
Author Response
Response to Reviewer 2 Comments
The association between parenting stress, positive reappraisal coping, and quality of life in parents with Autism Spectrum Disorder (ASD) children: An Analysis Review
Dear reviewer,
First of all, we would like to thank you for your constructive inputs for our manuscript. It has helped us to improve the manuscript, and we clarify text that might be confusing for the reader. We tried to revise the text according to reviewers suggestion. Hopefully, it will be answering the points raised by yours. In the response for the reviewer comment we repeat your comments, which point by point. Your comments are followed by our response in red color and the direction associated with changes made in the text. You will find that we adapted the text in all cases. We hope that you will appreciate the new version and hope to hear from you soon.
Thank you for your consideration of this manuscript. I look forward to hearing from you.
Sincerely,
Assoc. Prof. Mein-Woei Suen, Ph.D.
Department of Psychology, College of Medical and Health Science, Asia University, Taiwan blake5477@yahoo.com.tw or blake@asia.edu.tw.
Point 1: This ms presents a review of the literature over the last 10 years on parental stress, coping strategies, and quality of life. The ms is generally well written and the methods employed in the review process appear to meet PRISMA Standards. Figure 1 is appropriate and provides adequate information regarding the literature search process. Use of Kappa for interobserver agreement was also an appropriate method.
Overall, my recommendation is to publish the paper with minor revision. Those revisions include:
“Children with autism” is the preferred term for this population. An alternative, not using person-first language would be “autistic children.”
Response 1: We thank the reviewer for the constructive comments to improve this manuscript, we revised our article base on your suggestion.
Point 2: Is it “ineffective use of coping strategies” or lack of knowledge of coping strategies and/or lack of resources to use coping strategies? This would be an appropriate addition to the discussion and interpretation of the results.
Response 2: Thanks for your suggestion, we have revised it. Please, see line 387 and 397 in the grey marks.
Point 3: Table 3 does not seem to be necessary. Since all studies included all criteria (i.e., 100%), a note in the results section seems sufficient.
Response 3: Thanks for your suggestion, we have deleted table 3 from the text.
This manuscript is a resubmission of an earlier submission. The following is a list of the peer review reports and author responses from that submission.
Round 1
Reviewer 1 Report
Manuscript ID: Healthcare-1419126
General comments:
This systematic review focuses on summarizing recent literature exploring stress and coping in parents of children with ASD. The paper’s overall focus is important: dealing with children with severe ASD imposes substantial stress and psychological distress, especially for parents lacking access to social support and institutional resources. Exhausted parents sometime end up adopting counterproductive coping strategies that may lead to impaired child development, maladjusted relationships, and significant emotional distress. We do need more high quality data on linkages between parental stress, coping, and quality of life outcomes and a focused review of literature could provide important insights. Unfortunately, the manuscript suffers from a number of flaws that severely impair its value. Some of these issues are described below:
Specific comments:
- Please review the description on page 4 lines 156—171 thoroughly. It seems inconsistent with figure 1 in some respects (e.g., is it 12 or 15 duplicates?). Also, it is not clear failure of which inclusion criteria were responsible for excluding 4734 articles at the second stage (between screening and eligibility). Clarify the order in which the inclusion/exclusion criteria were applied.
- Table 2 is referred to in the manuscript but is missing. Please supply the missing table in the manuscript.
- The application of JBI evaluation scheme to the quality appraisal of the chosen studies raises some red flags. Typically such applications generate significant variation in appraisal scores along critical dimensions of study quality. However, it seems the authors found no such variation among the 29 papers selected for final review. Authors need to provide a more complete description of the process by which quality scores were assessed, clear definitions of critical dimensions of the quality appraisal scheme, and any inter-rater validation process used by authors to resolve conflicts. Authors also need to provide a clear justification for lack of variation among selected studies in quality appraisal results. Part of the issue may relate to their choice of exclusion criteria: their decision to review only crossectional designs is somewhat puzzling; experimental designs generally yield higher confidence that the underlying the relationships are truly causal.
- The discussion of the results is at many places incoherent and confusing and marred by numerous errors of grammar and syntax (inappropriate capitalization, puzzling insertion of “?” signs throughout the text, spelling errors, unnecessary text fragments, duplicate words, wrong tense etc.,). It seems the manuscript did not undergo a thorough copyediting process prior to circulation for review. The mechanics of many sentences is so obtuse that it is challenging to infer the authors’ intent (e.g., page 8, lines 292-297; page 15, lines 402-407). Authors’ summary of key findings also suffers from repetitiveness and excessive wordiness and inconsistent usage of terms (e.g., parents of ASD are variously described as ASD parents or ASD mothers or parental ASD). Please thoroughly review the manuscript to weed out these errors.

Reviewer 2 Report
First, I would like to say that I am very thankful to have the opportunity to read this study. The suggestions given in this document are intended to improve your work. The same feedback document will be given to both editors and authors.
I would like to recommend authors to carefully re-read the journal's instructions for authors, since, among other things, everything they include in supplementary material should be included in the paper itself. There are other issues regarding this. Please also check that you have used the references according to these guidelines.
https://www.mdpi.com/journal/healthcare/instructions
On the other hand, you attach a favorable document from an ethics committee, which is surprising since this type of research is done with papers, it does not involve human experimentation, so it is not necessary.
Abstract:
- According to the journal’s instructions: “The abstract should be a total of about 200 words maximum. The abstract should be a single paragraph and should follow the style of structured abstracts, but without headings”.
Introduction section:
- Line 47: Explain all acronyms the first time they are used, e.g., ASD, QoL.
- Line 46: The autism is a condition… better a disorder. To say that it is problematic is an opinion, please describe the symptomatology without making comments that may be subjective.
- Please expand the main symptomatology of autism. I also remind you that not all of them have behavioural problems.
- Line 50: “...are not only primary nurses in their life” What does this mean?
- Respectful language should be used throughout the text. The WHO recommends the use of people with disabilities, so use children with autism.
- Although the information given in this section is interesting, it should be restructured. Think that this section should help readers understand the main variables of your work and get an idea of what you intend to do. Make sure this is done. Also, there are research questions in the middle of the text, which is confusing, and the research objective should be clear enough to close the section.
Methods section:
- Although the title talks about a review analysis, when you get to the methodology it already talks about PRISMA statement, then the work must follow this statement.
- This section is complex to read. I attach several readings to help you improve the structure of the section. The diagram should follow the 2020 actualization.
- http://prisma-statement.org/PRISMAStatement/PRISMAStatement
- https://www.revespcardiol.org/en-the-prisma-2020-statement-an-articulo-S1885585721002401
- Examples:
- https://www.mdpi.com/2227-9032/9/8/1075/htm
- https://www.mdpi.com/2227-9032/9/8/1047/htm
- https://www.mdpi.com/2227-9032/9/7/889/htm
- Was the search register on PROSPERO o similar?
- Which tool was used to extract the information? An excel sheet, Covidence, Parsifal? It is not clear.
- What about the risk of bias assessment?
- What about the effect measures?
- I would like the authors to provide the PRISMA checklist, I do not see sufficient evidence that the study followed the statement.
Results section:
- There are no tables 1 and 2 in the manuscript.
Conclusion section:
- The conclusion is not really a conclusion, but a continuation of the discussion. The conclusion should only summarise the main findings of the study and answer the research question. Maybe part of the information given should be including in discussion.
Being a systematic review, I cannot understand all this information....
- “Institutional Review Board Statement: “The study was conducted according to the guidelines of the Declaration of Helsinki and approved by health research ethics committee the University of Muhammadiyah Malang No.E.5. a/038/KEPK-UMM/III/2021.
- Informed Consent Statement: Informed consent was obtained from all subjects involved in the study. Written informed consent has been obtained from the participants to publish this paper
- Acknowledgments: The authors would like to thank all the participants who have contributed to this study.”
Round 2
Reviewer 2 Report
Thanks to the authors for this new version, in which I can see that they have made a great effort. I indicate in red colour those recommendations from round 1 that have not been addressed and that I believe should be considered.
- Respectful language should be used throughout the text. The WHO recommends the use of people with disabilities, so use children with autism or ASD, not autistic.
- Although the information given in this section is interesting, it should be restructured. Think that this section should help readers understand the main variables of your work and get an idea of what you intend to do. The introduction needs another turn, it is more like a discussion. It is correct to mention some relevant research, but, are the authors sure that they have clearly explained what autism is, what causes stress to families, what is parental stress and its characteristics, quality of life and the Possitive Reappraisal Coping?
Congratulations.
Author Response
Response to Reviewer 2 Comments
The association between parenting stress, positive reappraisal coping, and quality of life in parents with Autism Spectrum Disorder (ASD) children: An Analysis Review
Dear reviewer,
Thanks for your guidance and constructive inputs for our manuscript. It has really helped us to improve the manuscript, and we clarify text that might be confusing for the reader. We tried to revise the text according to your suggestion. Hopefully, it will be answering the points raised by yours. We hope that you will appreciate the new version and hope to hear from you soon.
Thank you for your consideration of this manuscript. I look forward to hearing from you.
Sincerely,
Assoc. Prof. Mein-Woei Suen, Ph.D.
Department of Psychology, College of Medical and Health Science, Asia University, Taiwan blake5477@yahoo.com.tw or blake@asia.edu.tw.
Point 1: Respectful language should be used throughout the text. The WHO recommends the use of people with disabilities, so use children with autism or ASD, not autistic.
Response 1: We thank the reviewer for the constructive comments to improve this manuscript, we revised the word autistic with autism in all manuscripts.
Point 2: Although the information given in this section is interesting, it should be restructured. Think that this section should help readers understand the main variables of your work and get an idea of what you intend to do. The introduction needs another turn, it is more like a discussion. It is correct to mention some relevant research, but, are the authors sure that they have clearly explained what autism is, what causes stress to families, what is parental stress and its characteristics, quality of life and the Possitive Reappraisal Coping?
Response 2: Thanks for your constructive suggestion. We have restructured the information in the introduction based on your suggestion (Please see introduction with yellow mark)